# Combined Effect of Diosgenin Along with Ezetimibe or Atorvastatin on the Fate of Labelled Bile Acid and Cholesterol in Hypercholesterolemic Rats

**DOI:** 10.3390/ijerph16040627

**Published:** 2019-02-20

**Authors:** Alejandro Marín-Medina, Gonzalo Ruíz-Hidalgo, Jorge L. Blé-Castillo, Alma M. Zetina-Esquivel, Rodrigo Miranda Zamora, Isela E. Juárez-Rojop, Juan C. Díaz-Zagoya

**Affiliations:** 1División Académica de Ciencias de la Salud, Universidad Juárez Autónoma de Tabasco (UJAT), Villahermosa C.P. 86100, Tabasco, Mexico; stat5A@hotmail.com (A.M.-M.); jblecastillo@hotmail.com (J.L.B.-C.); almamileira@hotmail.com (A.M.Z.-E.); 2Universidad de Guadalajara, CUCS, Guadalajara C.P. 44340, Jalisco, Mexico; 3Hospital Angeles de Villahermosa, Villahermosa C.P. 86035, Tabasco, Mexico; derma.grh@gmail.com; 4Facultad de Medicina, División de Investigación, Departamento de Bioquímica, Universidad Nacional Autónoma de México, Ciudad Universitaria, Ciudad de México C.P. 04510, Ciudad de México, Mexico; mzr@bq.unam.mx

**Keywords:** cholesterol, diosgenin, atorvastatin, taurocholic acid, ezetimibe

## Abstract

We analyzed the effect of diosgenin, administered with atorvastatin or ezetimibe, on the fate of ^3^H(G)-taurocholic acid or 26-^14^C-cholesterol in hypercholesterolemic rats. Male Wistar rats received a hypercholesterolemic diet (HD), HD + atorvastatin (HD+ATV), HD + ezetimibe (HD+EZT), HD + diosgenin (HD+DG), HD+ATV+EZT, or HD+ATV+DG for 40 days. We also included a control normal group (ND). The labelled compounds were administered on day 30. The animals were placed in metabolic cages for daily feces collection. At day 40 the rats were sacrificed. Lipid extracts from blood, liver, spinal cord, testicles, kidneys, epididymis, intestine, and feces were analyzed for radioactivity. Cholesterol activity was the highest in the liver in HD rats. DG diminished one half of this activity in HD+DG and HD+ATV+DG groups in comparison with the HD group. HD+ATV rats showed four to almost ten-fold cholesterol activity in the spinal cord compared with the ND or HD rats. Fecal elimination of neutral steroids was approximately two-fold higher in the HD+DG and HD+ATV+DG groups. Taurocholic acid activity was four to ten-fold higher in HD+DG intestine as compared to the other experimental groups. Taurocholic activity in the liver of HD and HD+DG groups was two and a half higher than in ND. Our results show that the combination of DG and ATV induced the highest cholesterol reduction in the liver and other tissues.

## 1. Introduction

Dyslipidemia has a crucial role in the pathophysiology of cardiovascular disease secondary to the development of atherosclerosis [1]. Cholesterol is a steroid widely distributed in cell membranes, and is one of the main bile constituents. Cholesterol synthesis is performed primarily in the liver, while the kidneys, intestine, and adrenal glands play a secondary role [2]. Although the life-style change is the method of first choice for dyslipidemia control, nowadays statins are the more commonly used group of drugs for lowering blood cholesterol levels. Atorvastatin (ATV) competitively inhibits the enzyme 3-hydroxy-3-methylglutaryl CoA reductase (HMG CoA reductase), and thus regulates endogenous cholesterol biosynthesis [3]. On the other hand, ezetimibe (EZT) is an inhibitor of intestinal cholesterol absorption, specifically at the level of conveyor Niemann–Pick C1-like protein 1 (NPC1L1) through the intestinal wall, without interfering with the absorption of triacylglycerols, liposoluble vitamins, fatty acids, and bile acids [4]; in mice treated with EZT, a 92–96% decrease in the absorption of cholesterol has been observed [5].

Diosgenin (DG) is the aglycone of saponins of *Dioscoreaceae* family plants; it can modify some metabolic sequences of cholesterol [6]. Diosgenin administration can accelerate the conversion of cholesterol into bile acids in animal models and has an anti-inflammatory effect due to its structural similarity to the estrogens [7]. Diosgenin has been proposed as an active therapeutic tool in several diseases (diabetes mellitus, dyslipidemia, inflammatory processes) [8]. It can induce the expression of vascular endothelial growth factor (VEGF-A) in osteoblasts (angiogenesis) [9], and it has recently been found that it plays a vital role in the metabolism of glucose and lipids [10].

The reverse transport of cholesterol plays a significant role in carrying excess cholesterol from the tissues to the liver; this carrier action is continued with biliary excretion through transintestinal cholesterol excretion (TICE) [11]. It is considered that improving the efflux of cholesterol from HDL (high-density lipoproteins) particles reduces the risk of cardiovascular diseases since the risk is inversely related to the efflux of cholesterol. The proteins that play a central role in the efflux of cholesterol in organs are the ABC transporters [11]. TICE plays a crucial role in the excretion of dietary and biliary cholesterol; this route allows the direct elimination of cholesterol through the enterocyte [12]. It is postulated that this pathway could have a compensatory role when there is dysfunction in the reverse transport of cholesterol, although it seems that this role may be conditioned to other factors [13]. It has been proven that ATV, EZT, and DG can modify the expression of various proteins related to the transport and efflux of cholesterol through different mechanisms [14,15,16].

The aim of this study was to analyze the effect of DG, ATV, and EZT in monotherapy or in combination on the fate of ^3^H(G)-taurocholic or 26-^14^C-cholesterol administered to hypercholesterolemic rats.

## 2. Materials and Methods 

### 2.1. Animals and Diets

Male albino rats (Wistar) weighing 200–250 g were fed powdered Harlan chow containing 18% protein, 6.5% fat, and 3.5% fiber. The care of the animals was in accordance with the Mexican norm for animal use in laboratory NOM-062-ZOO-1999.
ND: Normal dietHD: Hypercholesterolemic diet (2% cholesterol, 0.06% sodium deoxycholate)HD+ATV: HD + atorvastatin 0.09 mg/kgHD+EZT: HD + ezetimibe 1.66 mg/kgHD+DG: HD + diosgenin 5%HD+ATV+EZT: HD + atorvastatin 0.09 mg/kg+ ezetimibe 1.66 mg/kgHD+ATV+DG: HD + atorvastatin 0.09 mg/kg+ diosgenin 5%


The doses of ATV and EZT were selected according to therapeutic doses in humans. These drugs were ground in a pestle and mixed with the ground food. Diets were freshly prepared each day with grinded food and were given over the course of 40 days, and on day 30 the animals received by a single intraperitoneal injection of labelled substances. The purpose of this study was to determine in hypercholesterolemic rats was to determine the distribution of the labelled compounds without the initial intestinal absorption but keeping the participation of the enterohepatic cycle. Accordingly, the rats were injected intraperitoneally with ^3^H(G)-taurocholic acid 1 × 10^5^ disintegrations per minute (dpm) in 200 µL of ethanol/saline solution (1:1 v/v) or 26-^14^C-cholesterol (1 × 10^6^ dpm) in the same vehicle [17,18]. For each treatment at least six animals were included. Animals were maintained in individual metabolic cages and the feces were collected every day during 10 days. On day 40, after 8 hours of fasting, animals were sacrificed and blood, liver, small intestine, spinal cord, kidneys, testicles, and epididymis were harvested.

### 2.2. Test Compounds

Diosgenin and sodium deoxycholate were purchased from Sigma Chemical Co (St Louis) and were 95% pure. Ezetimibe (10-mg tablets) was from Shering-Plough. Atorvastatin (20-mg tablets) was from Pfizer Labs. ^3^H (G)-taurocholic acid was purchased from Perkin Elmer Life and Analytical Sciences (Boston). 26-^14^C-cholesterol was purchased from Dupont NEN products (Boston). Other reactants of analytical grade were purchased from Merck (Mexico). The radioactive compounds were used in accordance with the *Reglamento General de Seguridad Radiológica y Norma Oficial Mexicana NOM-012-STPS-1993*.

### 2.3. Biochemical Parameters

Glucose, cholesterol, and triacylglycerols in blood serum were analyzed using a Clinical Chemistry System from Random Access Diagnostics. 

The lipid extract of the liver, spine cord, kidney, testicle, epididymis, serum, and feces was obtained by a modified Folch method [19], employing chloroform–methanol (2:1 v/v). Cholesterol and triacylglycerol content of the liver was evaluated in the extracts using enzymatic colorimetric methods. The extract was also used to determine the ^14^C activity after the labeled cholesterol injection or the ^3^H activity after the labeled taurocholic acid injection. Radioactivity was measured using a Beckman LS 6500 Scintillation counter.

### 2.4. Statistical Analysis

Data are expressed as the mean ± SE of six values. Significance of differences was determined for each group of values by one-way ANOVA followed by the Student–Newman–Keuls test; values *p* < 0.05 were considered significant. 

## 3. Results

### 3.1. Biochemical Values

Serum glucose in HD treatment did not modify glucose levels with respect ND, however the HD+DG and HD+EZT groups showed lower values (*p* < 0.05). Serum cholesterol showed a significant increase in HD animals compared with ND (*p* < 0.05) (Table 1). Nevertheless, the HD+DG, HD+EZT and HD+ATV+EZT groups had significantly lower values (*p* < 0.05) in comparison with the HD group (Table 1). 

Total cholesterol in the liver in HD was significantly higher than in ND animals (*p* < 0.05). In the HD+ATV group a significant decrease was observed with respect to the HD group (*p* < 0.05). Hepatic triacylglycerols levels were higher in the HD group than in ND group (*p* < 0.05). The other groups had significant lower values compared with the HD group (*p* < 0.05), except the HD+ATV+EZT group, which showed a moderate decrement (*p* < 0.05). 

### 3.2. Destiny of Injected Labelled Cholesterol or Cholic Acid

The serum level of the radiotracers was generally low after 10 days of injection either of taurocholic acid or cholesterol. Taurocholic acid ^3^H activity in ND group was 91.06 ± 14.9 dpm/mL and there were no significant differences with the other groups. The cholesterol ^14^C activity in serum was also low (500 dpm/mL).

### 3.3. Distribution of ^14^C-cholesterol in the Liver, Spinal Cord, Kidney, Testes, and Epididymus

The tissue distribution of 26-^14^C-cholesterol in the rats is shown in Table 2. The hepatic and kidney tissues had the highest ^14^C activity levels in ND (689 ± 38.81 and 1091 ± 127.37 dpm/0.5 g, respectively). The HD group even had higher significant activity in the liver and spinal cord with respect to ND (*p* < 0.05). The hepatic and epididymal tissues had lower activity in HD+DG and HD+ATV+DG rats than the HD rats (*p* < 0.05), and the other four groups showed activities between HD and ND values. The ^14^C activity in the liver of HD+EZT was the highest of all groups but no significant difference was observed with respect to ND. The spinal cord ^14^C activity in the ND, HD, and HD+EZT groups corresponded approximately to one half of the activity in their respective liver tissue; besides, in HD+DG and HD+ATV+DG the liver and the spinal cord had equivalent activity. However, HD+ATV spinal cord had more than two folds of the liver activity after the same treatment and almost nine times the activity of ND (*p* < 0.05) (Table 2). 

The kidney in the HD group showed a significant lower activity as compared to the ND group (*p* < 0.05). In the HD+ATV+EZT group the kidney ^14^C activity surpassed that of the liver (*p* < 0.05) (Table 2). In addition, testicles in all groups presented the lowest ^14^C activity, with no statistical differences between groups (Table 2). 

### 3.4. Taurocholic Acid in the Liver, Intestine, and Serum

Evaluation of ^3^H activity of taurocholic acid in the HD liver increased significantly with respect to ND (*p* < 0.05). The liver ^3^H activity values were significantly lower in all treatments compared with HD (*p* < 0.05) (Table 3). On the other hand, a significant higher ^3^H activity was observed in HD+DG intestine in comparison with that observed in ND and HD (*p* < 0.05). In addition, HD+EZT intestine showed a significant lower ^3^H activity with respect to the HD group (*p* < 0.05) (Table 3). Serum ^3^H activity was very low in all groups, with no significant differences between the ND group and the other groups.

### 3.5. Fecal Elimination of Neutral Steroids

The ^14^C activity in feces was moderately higher in HD compared with ND, but elimination was even higher in HD+DG and HD+ATV+DG (*p* < 0.05). Neutral steroid elimination was significantly lower in HD+EZT, HD+ATV and HD+ATV+EZT than in HD group (*p* < 0.05) (Table 4). 

### 3.6. Fecal Elimination of Acidic Steroids

Elimination of taurocholic acid measured by the ^3^H activity in feces showed similar values in all groups but HD+DG, in which the activity was significantly lower (4.82 ± 0.21 × 10^3^ dpm/0.5 g) than that of ND (6.37 ± 0.63 × 10^3^ dpm/0.5 g) (Table 4). 

## 4. Discussion

The particular objectives of the study were: (1) to analyze the effect of DG, ATV, and EZ in monotherapy or combined on the serum levels of cholesterol, glucose and triglycerides, as well as on the hepatic cholesterol and triglyceride concentration; (2) to analyze the concentration of cholesterol labeled in the liver, spinal cord, kidney, testis, epididymis, and serum in the different groups; (3) to analyze the enterohepatic cycle of bile acids by marking with taurocholic acid; and (4) to determine the effect of these substances on the fecal excretion dynamics of acidic and neutral steroids.

### 4.1. Biochemical Parameters

Our findings showed that all treatments maintained cholesterol and triacylglycerols lower than in hypercholesterolemic rats without treatment, especially after DG administration. It is well known that diosgenin shows activity on lipid metabolism, although the mechanism is still not clear. Interestingly, diverse authors proposed that diosgenin intervenes on the pathways regulated by sterol regulatory element-binding proteins (SREBPs), elements that regulate the expression of some essential enzymes of lipid metabolism, such as acetyl-CoA carboxylase [8,20] and HMG-CoA reductase [21,22]. In addition to this hypolipemiant effect, diosgenin participates in glucose metabolism, reduces oxidative stress, supports hepatoprotective enzymes in blood plasma and liver, and lowers the expression of caspases, preventing apoptosis in cells subjected to hyperglycemic stress [23,24]. These reports are in agreement with our findings that serum glucose had a lower value in rats that received DG (Table 1). 

ATV exhibits hypocholesterolemic activity attributed to the inhibition of the enzyme HMG-CoA reductase. However, other mechanisms have been proposed for this drug; ATV indirectly modifies the transport of cholesterol by changing the pathway of proteins that interact with SREBP [25], inhibiting the efflux of cholesterol by means of change in expression of ATP-binding cassette protein A1 (ABCA1) responsible for transport of xenobiotics and cholesterol [26]. In this sense, for the combined effect of AVT and DG in HD rats suggest that DG has more than one mechanism of action on cholesterol, and regulates the expression of molecule that plays an essential role in the metabolism of lipids and carbohydrates. In another hand, ATV is a blocker of cholesterol synthesis through the synthesis at the HMG-CoA reductase enzyme. These effects together possibly involve different signaling pathways of lipid metabolism [23,24,25]. 

EZT has the effect of lowering blood cholesterol by blocking Niemann–Pick C1-like 1 protein (NPC1L1) and produces a reduction of low-density lipoprotein-cholesterol (LDL-C) in plasma, affecting the reverse transport of cholesterol, since it has been proposed that EZT increases the activity of the ABCG5/G8 transporter, which is primary to the intestine, temporarily increasing the efflux of cholesterol [27]. Our data show that the combination of ATV with DG or EZT decreases cholesterol and triacylglycerols in HD rats. 

### 4.2. Distribution of Labelled Cholesterol in Several Organs

In all organs except the kidney, an increase in activity was observed in the HD group compared to the ND group. Cholesterol-enriched diets damage myeloid cells, endothelial cells, lymphocytes, and central nervous system (CNS) cells. Our results show that the liver and kidney of ND rats concentrated the major ^14^C-cholesterol activity (Table 2) and that the rats with HD had more than 50% increase in activity in the liver, testis, and spinal cord as compared to the same tissues in the ND rats. Little is known about effect of DG on the central nervous system [28,29]. 

We observed that ATV increased the activity of cholesterol 26-^14^C detected in the spinal cord by almost 10 times its value in relation to rats with ND and almost 4 times in relation to rats with HD. Molecular mechanisms related to CNS cholesterol homeostasis are not entirely clear. It has been found that a prolonged ingestion of statins changes cholesterol homeostasis at the CNS and in this way they could affect myelination [30]; besides, ATV is a lipolytic compound that passively crosses the blood–brain barrier, increasing its concentrations in the nervous tissue [31] Other researchers propose that statins probably change the expression of certain genes in CNS, especially transporters organic anion transporting polypeptides (OATPs). In the case of the epididymis we observed that DG alone or in combination showed significant differences with respect to ND and HD, and the ATV showed significant differences alone and in combination with respect to ND and HD, so it seems that DG reduces the transport of cholesterol towards the epididymis. It is known that there may be variations in the transport of cholesterol towards the epididymis under certain conditions; for example, during spermatogenesis, cholesterol transport is increased [32]. The mechanisms that regulate the transport of cholesterol to the epididymis are not fully understood, however it is known that the ABCA1 transporter is the main transporter of cholesterol in this tissue [33] so DG and ATV could modify the expression of this protein and therefore change the amount of cholesterol that is concentrated in epididymis. In the kidney, a significant difference was observed in the EZT group compared to HD group. EZT increases the concentration of cholesterol in the kidney compared to HD, but shows similar concentrations to ND, so apparently with high concentrations of cholesterol it is distributed mainly to the liver, suggesting that under these conditions EZT could have a role in the regulation of cholesterol in the kidney [32,33].

### 4.3. Taurocholic Acid in the Liver, Intestine, and Serum and Fecal Elimination of Neutral and Acidic Steroids

The main sterol of animal origin is cholesterol. Endogenous and exogenous cholesterol are precursors of neutral fecal steroids. Cholesterol in bile, in intestinal secretions, and in sloughed mucosal cells also contribute as precursors of neutral fecal steroids. Bile acids are also excreted in feces; under normal conditions they represent a complex mixture of metabolites formed by microbial enzymes. Previous studies showed that 73% of bile acids were found in the intestine of rats treated with diosgenin [34]. In the same way, we observed a higher activity of ^3^H(G)-taurocholic acid in the intestine of rats treated with DG. We suggest that DG increases the output of bile acids from hepatocytes into the bile and increases its concentration in the intestine, possibly by regulating the expression of genes encoding the proteins mentioned above [35,36,37].

The main metabolic destination of cholesterol is the synthesis of bile acids. It has been observed that EZT decreases the fecal excretion of neutral sterols, but apparently does not modify the excretion of acid sterols [38], whereas it has been reported that ATV stimulates the excretion of cholesterol at the intestinal level and increases the synthesis of bile acids [14,15,39]. These reports are in part similar to our results (Table 4 and Table 3, respectively). Previously, NPC1L1 was identified as the protein involved in the intestinal absorption of cholesterol and as a therapeutic target of ezetimibe [24]. In addition, the ABCG5/G8 protein participates in the return of absorbed sterols to the intestinal lumen, as a regulatory mechanism to prevent sterol overload [36]. It has been reported that the expression of this protein is modified in response to EZT [40], and thus EZT could be reducing the amount of neutral steroids in feces and the concentration of bile acids into the intestine by modifications in the expression of this protein and in some way redistributing cholesterol to the kidney.

In the same manner, we observed increased fecal excretion of neutral sterols in DG-treated rats compared with ND rats. Regarding the hypocholesterolemic effect observed in rats treated with diosgenin, it has been proposed that this compound promotes fecal excretion of sterols by biliary cholesterol secretion and inhibition of intestinal absorption [29,37,41], but this effect seems to be independent of the cholesterol absorption mediated by NPC1L1 [24]. 

On the other hand, ATV also modifies the metabolism of bile acids by increasing the expression of enzyme 7-α-hydroxylase (this enzyme regulates the synthesis of bile acids), inhibiting the expression of FXR (farnesoid X-activated receptor) [42,43]. These observations suggest that ATV can change the homeostasis of bile acids by modifying their concentration in the intestine but without affecting their serum and liver levels [42,43]. Apparently DG, ATV, and EZT in monotherapy or in combination do not seem to have an effect on the excretion dynamics of acid steroids.

## 5. Conclusion

In conclusion, the present study shows that DG, ATV, and EZT alone or in combination have important roles in controlling lipid metabolism. Besides, DG and EZT improve the intestinal absorption of glucose and cholesterol. However, ATV increased the distribution of marked cholesterol in the spinal cord, possibly by different mechanisms that alter cholesterol homeostasis in the CNS. In addition, EZT seems to increase the distribution of cholesterol to the kidney and decrease the fecal excretion of neutral steroids. However, these findings require additional studies in animal models to evaluate the participation of ATV, EZT, and DG in the expression of genes associated with transport and cholesterol homeostasis.

## Figures and Tables

**Table 1 ijerph-16-00627-t001:** Effect of atorvastatin, ezetimibe, and diosgenin or their combination on serum and liver biochemical parameters of hypercholesterolemic rats.

	ND	HD	HD+ATV	HD+DG	HD+EZT	HD+ATV+DG	HD+ATV+EZT
Serum (mg/dL)							
Glucose	96.4 ± 6.90	95.00 ± 5.75	90.20 ± 9.49	72.67 ± 1.7 *^,#^	83.33 ± 4.68 *,^#^	85.50 ± 4.45	97.00 ± 3.42
Cholesterol	76.2 ± 3.29	161.5 ± 11.8 *	119.2 ± 27.37	71.67 ± 9.64 ^#^	95.83 ± 13.62 ^#^	136.0 ± 12.52	81.83 ± 9.03 ^#^
Triacylglycerols	58.60 ± 3.29	93.57 ± 10.26 *	73.40 ± 10.3	50 ± 6.43 ^#^	65.67 ± 5.25	59.40 ± 4.49 ^#^	65.33 ± 9.03
Liver (mg/g)							
Cholesterol	2.60 ± 0.35	17.39 ± 3.55 *	3.717 ± 0.53 ^#^	13.65 ± 0.48	18.34 ± 2.5	11.25 ± 0.68	17.12 ± 1.49
Triacylglycerols	24.61 ± 1.34	50.27 ± 1.01 *	6.79 ± 1.8 ^#^	2.62 ± 0.4 ^#^	5.32 ± 1.29 ^#^	4.16 ± 0.62 ^#^	17.58 ± 0.63 ^#^

Values represent the mean ± SEM (*n* = 6). * Statistically different from normal diet (ND), ^#^ Statistically different from hypercholesterolemic diet (HD), (*p* < 0.05), by one-way ANOVA followed by Student–Newman–Keuls. HD+ATV: hypercholesterolemic diet + atorvastatin; HD+EZT: hypercholesterolemic diet + ezetimibe; HD+DG: hypercholesterolemic diet + diosgenin; HD+ATV+EZ: hypercholesterolemic diet + atorvastatin + ezetimibe; HD+ATV+DG: hypercholesterolemic diet + atorvastatin + diosgenin; dpm: disintegrations per minute.

**Table 2 ijerph-16-00627-t002:** Body distribution of 26-^14^C-cholesterol in rats with a cholesterol-rich diet that received DG, ATV, EZT, or their combinations.

26-^14^C-Cholesterol (dpm/0.5 g Tissue or dpm/L Serum)	Liver	Spinal Cord	Kidney	Testis	Epididymis	Serum
ND	689 ± 38.81	243 ± 99	1091 ± 127.37	114.0 ± 54.26	259 ± 37.57	360 ± 72.1
HD	1574 ± 284.2 *	669.4 ± 181 ^,#^	615.8 ± 150 *^,#^	297.7 ± 27.2 *^,#^	483.7 ± 93 *^,#^	517 ± 17 ^#^
HD+ATV	905.4 ± 94.67 ^#^	2330 ± 859.4 *	492.7 ± 98.75 ^#^	158.7 ± 26.02 ^#^	212.4 ± 44.74 ^#^	192.6 ± 13.01 *^,#^
HD+DG	590.9 ± 157.3 ^#^	566.7 ± 128.3	749.7 ± 164.2	177.4 ± 26.11 ^#^	90.8 ± 20.3 *^,#^	260 ± 78.87 ^#^
HD+EZT	1599 ± 140.0	584 ± 87.24 *	1085 ± 208 ^#^	261.5 ± 45.60 *	309.6 ± 145.8	378.2 ± 34.10 ^#^
HD+ATV+DG	521.0 ± 82.7 ^#^	436.1 ± 82.69	263.9 ± 80 *^,#^	126.5 ± 27.74 ^#^	80.16 ± 6.74 *^,#^	135.6 ± 3.07 *^,#^
HD+ATV+EZT	688.7 ± 61.55 ^#^	295.2 ± 44.35 ^#^	866.5 ± 93.84	177.6 ± 8.432 ^#^	206.6 ± 19.80 ^#^	382.9 ± 45.67 ^#^

Data are the mean ± SEM of disintegrations per minute (dpm) in six animals. * Statistically different from normal diet (ND), ^#^ Statistically different from hypercholesterolemic diet (HD), (*p* < 0.05), by one-way ANOVA followed by Student–Newman–Keuls. HD+ATV: hypercholesterolemic diet + atorvastatin; HD+EZT: hypercholesterolemic diet + ezetimibe; HD+DG: hypercholesterolemic diet + diosgenin; HD+ATV+EZT: hypercholesterolemic diet + atorvastatin + ezetimibe; HD+ATV+DG: hypercholesterolemic diet + atorvastatin + diosgenin.

**Table 3 ijerph-16-00627-t003:** Body distribution of ^3^H(G)-taurocholic acid in hypercholesterolemic rats that received DG, ATV, EZT or their combinations.

^3^H(G)-taurocholic Acid	ND	HD	HD+ATV	HD+DG	HD+EZT	HD+ATV+DG	HD+ATV+EZT
Liver (dpm/0.5 g of tissue )	152.6 ± 27.7	525.6 ± 38.29 *	312.6 ± 41.72 ^#^	415.3 ± 25.97 ^#^	173.1 ± 51.35 ^#^	290.2 ± 31.83 ^#^	121.5 ± 16.29 ^#^
Intestine (dpm/0.5 g of tissue)	93.28 ± 14.23	184.4 ± 27.67 *	270.8 ± 79.6	1143 ± 283.2 *^,#^	76.83 ± 23.16 ^#^	240.6 ± 0.68	156.9 ± 55.01
Serum (dpm/mL)	91.06 ± 14.85	77.54 ± 15.65	68.58 ± 5.49	92.06 ± 17.54	69.38 ± 17.54	65.52 ± 5.046	79.27 ± 15.23

Data are the mean ± SEM of disintegrations per minute (dpm) in six animals. * Statistically different from normal diet (ND), ^#^ Statistically different from hypercholesterolemic diet (HD), (*p* < 0.05), by one-way ANOVA followed by Student–Newman–Keuls. HD+ATV: hypercholesterolemic diet + atorvastatin; HD+EZT: hypercholesterolemic diet + ezetimibe; HD+DG: hypercholesterolemic diet + diosgenin; HD+ATV+EZT: hypercholesterolemic diet + atorvastatin + ezetimibe; HD+ATV+DG: hypercholesterolemic diet + atorvastatin + diosgenin.

**Table 4 ijerph-16-00627-t004:** Effect of atorvastatin, ezetimibe, diosgenin or their combinations on fecal 26-^14^C-cholesterol or ^3^H(G)-taurocholate elimination in hypercholesterolemic rats.

**AUC (dpm × 10^3^/0.5 g of Tissue)**	**ND**	**HD**	**HD+ATV**	**HD+DG**	**HD+EZT**	**HD+ATV+DG**	**HD+ATV+EZT**
**26-^14^C-cholesterol**	12.44 ± 0.97	17.61 ± 1.66 *	8.69 ± 1.16 ^#^	18.09 ± 1.44	7.2 ± 0.76 ^#^	20.92 ± 2.25	2.25 ± 1.44 ^#^
**^3^H(G)-taurocholic acid**	6.37 ± 0.63	5.70 ± 0.49	6.15 ± 0.16	4.82 ± 0. 21	6.16 ± 0.29	6.45 ± 0.39	5.77 ± 0.39

Data represents the AUC (area under curve) mean ± standard error of six rats of disintegrations per minute (dpm). * Statistically different from normal diet (ND), ^#^ Statistically different from hypercholesterolemic diet (HD), (*p* < 0.05), by one-way ANOVA followed by the Student–Newman–Keuls. HD+ATV: hypercholesterolemic diet + atorvastatin; HD+EZT: hypercholesterolemic diet + ezetimibe; HD+DG: hypercholesterolemic diet + diosgenin; HD+ATV+EZ: hypercholesterolemic diet + atorvastatin + ezetimibe; HD+ATV+DG: hypercholesterolemic diet + atorvastatin + diosgenin.

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
