# Peer review of "Combined Effect of Diosgenin Along with Ezetimibe or Atorvastatin on the Fate of Labelled Bile Acid and Cholesterol in Hypercholesterolemic Rats"

_ijerph, 2019, doi:10.3390/ijerph16040627_

Round 1

Reviewer 1 Report

Accept with minor revision.

Dear authors, 
I have reviewed the manuscript entitled "
Combined Effect of Diosgenin Along with Ezetimibe or Atorvastatin on the Fate of Labeled Bile Acid and Cholesterol in Hypercholesterolemic Rats". Your study is very valuable and important as it alert people and health care system about this kind of material. I appreciate the scientific quality of the experiment, data treatment and interpretation. In addition, the manuscript is well written, fluent and well addressed to the hypothesis the authors wanted to test. After some minor revisions which are proposed below, the manuscript is suitable for publication in this journal. I only have few minor comments; please find them listed below: 

1.       Dose selection criteria of tested drug must be included in material and method section. Moreover drugs (ATV, EZT and DG) used in the study are tablets, hence preparation of dosage and vehicle medium for administration in rats should be included and clarify the method of separation of excipients. If possible batch no. of drugs should also be included.

2.       It is not clear that whether labeled substance given on 30th day as a single dose or given for 10 consecutive days.

3.       Although HD+DG and HD+AVT alone and in combination show excellent results yet combination of HD+EZT are also good. Addition of another group HD+AVT+DG+EZT may be more promising in this study.

4.       Since cholesterol metabolism is complex process and study have multiple targets. It is suggested to add either working hypothesis chart or include a mechanistic pathway showing targets of study in discussion section.

5.       Probable mechanism of combined effect of HD+AVT+DG should be included in discussion.

6.       Page 4 line 141, small bowl should be replaced with intestine.      

7.       Page 5 line 154, “HD showed lower activity than ND-How it is possible?

8.       A small description about neutral steroids and acidic steroids should be included in discussion section.

9.       There are many small typological errors were noticed like-page 4 line 238; page 4 line 141, 143 and 144.

10.   Page 6 last para- discussion should be correlate with the results.

11.   Several references were found written in Upper case like reference no. 23, 31, 32 etc.

12.   Correct reference no. 1- Davis, K.    

Author Response

February 7, 2019

Carmen Ruiz, M.Sc.

IJERPH Editorial Office

Dear Ruiz:

Please find the revised version of our manuscript ID: ijerph-434373 entitled “Combined Effect of Diosgenin Along with Ezetimibe or Atorvastatin on the Fate of Labelled Bile Acid and Cholesterol in Hypercholesterolemic Rats” by Alejandro Marín-Medina, Gonzalo Ruíz-Hidalgo, Jorge L. Blé-Castillo, Alma M. Zetina-Esquivel, Rodrigo Miranda-Zamora, Isela E. Juárez-Rojop *, Juan C. Díaz-Zagoya *, which we are re-submitting to be considered for possible publication in Int. J. Environ. Res. Public Health; section: Health Behavior, Chronic Disease and Health Promotion.

We have carefully considered the reviewers comments and Editor´s remarks as follows:

Reviewer 1

Comments and Suggestions for Authors:  Accept with minor revision.

Dear authors,

I have reviewed the manuscript entitled "Combined Effect of Diosgenin Along with Ezetimibe or Atorvastatin on the Fate of Labeled Bile Acid and Cholesterol in Hypercholesterolemic Rats". Your study is very valuable and important as it alert people and health care system about this kind of material. I appreciate the scientific quality of the experiment, data treatment and interpretation. In addition, the manuscript is well written, fluent and well addressed to the hypothesis the authors wanted to test. After some minor revisions which are proposed below, the manuscript is suitable for publication in this journal. I only have few minor comments; please find them listed below:

1. Dose selection criteria of tested drug must be included in material and method section. Moreover, drugs (ATV, EZT and DG) used in the study are tablets, hence preparation of dosage and vehicle medium for administration in rats should be included and clarify the method of separation of excipients. If possible batch no. of drugs should also be included. This observation was included in the material and methods section (pag. 2, line 83-85)

2.It is not clear that whether labeled substance given on 30th day as a single dose or given for 10 consecutive days. We have included this point in the material and methods section (pag. 2, line 85 and 86)

3.Although HD+DG and HD+AVT alone and in combination show excellent results yet combination of HD+EZT are also good. Addition of another group HD+AVT+DG+EZT may be more promising in this study. We agree. We have included this point in the revised version of this manuscript.

4. Since cholesterol metabolism is complex process and study have multiple targets. It is suggested to add either working hypothesis chart or include a mechanistic pathway showing targets of study in discussion section. It was included in the first paragraph of the discussion (pag. 9. Line: 196-198). The particular objectives of the study were: a) to analyze the effect of DG, ATV and EZ in monotherapy or combined on the serum levels of cholesterol, glucose and triglycerides, as well as the hepatic cholesterol and triglyceride concentration; b) to analyze describe the concentration of cholesterol labeled in the liver, spinal cord, kidney, testis, epididymis and serum in the different groups; c) analyze the enterohepatic cycle of bile acids by marking with taurocolic acid; d) know the effect of these substances on the dynamics of fecal excretion of acidic and neutral steroids.

5.Probable mechanism of combined effect of HD+AVT+DG should be included in discussion. Discussion on this point has been included in the revised version (pag. 6, line: 218-223)

6. Page 4 line 141, small bowl should be replaced with intestine. Corrected

7. Page 5 line 154, “HD showed lower activity than ND-How it is possible?

Apparently in the HD group, the kidney was the only organ with a lower activity respect to ND, it seems than in situations where the concentration of cholesterol increases, this is distributed mainly to the liver. It is also known that the concentrations of cholesterol in the organs vary according to the metabolic needs, however this dynamic is not well known. On the other hand, in the group with EZT there was a noticeable increase in activity in the kidney, and it was observed that a decrease in the neutral sterols excretion was present suggesting that EZT seems to redistribute cholesterol to the kidney.

8. A small description about neutral steroids and acidic steroids should be included in discussion section. Discussion on this point has been included in the revised version (Section 4.3. Taurocholic acid in the liver, intestine and serum and Fecal elimination of neutral and acidic steroids)

9. There are many small typological errors were noticed like-page 4 line 238; page 4 line 141, 143 and 144. Corrected

10.Page 6 last para- discussion should be correlate with the results. We have included in the discussion section (page 7; line: 228-229)

11.Several references were found written in Upper case like reference no. 23, 31, 32 etc. We have modified this part.

12.Correct reference no. 1- Davis, K. Done

Best regards

Juan C. Diaz-Zagoya

Reviewer 2 Report

This manuscript needs a very deep revision to be reconsidered 

Minor revision:

-           many imprecisions as for example line 67-68: “The aim of this study was to analyze the combined effect of DG along with ATV or EZT”, but authors analyze the effect of the single drugs and various combinations among them. Correct also legends adding the effect of combinations of drugs.

-          line 78, 0.5% or 5% as then reported in line 80?

-          Line 147, add ”or normal diet”

-          Table 2, “a” underlined, what does it mean? Or ”,” before a letter?

Major revision:

Several mistakes about statistics if you consider what is written in the text respect to what is reported in tables. This makes very difficult to evaluate this manuscript. Just some example:

-          line 114,  for HD+EZT group  no statistics is reported in table 1

-          line 116, (p<0.0001, p<0.05 and p<0.0001; respectively)  data not correct if I look at the table (b for all the groups)

-          line 128, p<0.001 or 0.0001? it is not written in table what value is “a”; somewhere in the text  it is 0.0001

-          line 177, (p< 0.05) but in table there it is not “b”

-          etc, etc, etc…….

About Discussion:

-          too long because authors report several information which should be in introduction, not in this section

-          line 213, authors refer to fecal excretion in a wrong context (Biochemical parameters) and they talk about stimulation by diosgenin of cholesterol excretion but in their results I don’t find this  (table 4).

-          There are a lot of data not discussed. For example authors don’t discuss epididymis values that in some cases seem statistically significant. Then, line 229, “HD rats had 50% more activity in the liver, kidney and spinal cord than the same tissues in ND”…kidney? It doesn’t seem from table 2.

-          Then, I would compare data obtained for not labeled and labeled cholesterol that means discuss about the effect of drugs on cholesterol coming from synthesis and absorption (not labeled) with respect to the effect on cholesterol from just diet(labeled). For example, line 232-233, “DG diminished 26-14C-cholesterol activity in the liver and spinal cord”,  compare this with data in table 1 and you will not find the same, it would be nice to discuss this. Moreover data from spinal cord is not statistically relevant (see table 2)

-          Line 234 “ATV increased 26-14C-cholesterol activity detected in spinal cord”, please discuss why, in your opinion it is not the same for the combination ATV-EZT, considering also the effect of EZT alone.

-          Line 252,  “we observed increased fecal excretion of neutral sterols in DG treated rats”, it doesn’t seem from table 4

Author Response

February 7, 2019

Carmen Ruiz, M.Sc.

IJERPH Editorial Office

Dear Ruiz:

Please find the revised version of our manuscript ID: ijerph-434373 entitled “Combined Effect of Diosgenin Along with Ezetimibe or Atorvastatin on the Fate of Labelled Bile Acid and Cholesterol in Hypercholesterolemic Rats” by Alejandro Marín-Medina, Gonzalo Ruíz-Hidalgo, Jorge L. Blé-Castillo, Alma M. Zetina-Esquivel, Rodrigo Miranda-Zamora, Isela E. Juárez-Rojop *, Juan C. Díaz-Zagoya *, which we are re-submitting to be considered for possible publication in Int. J. Environ. Res. Public Health; section: Health Behavior, Chronic Disease and Health Promotion.

We have carefully considered the reviewers comments and Editor´s remarks as follows:

Reviewer 2

Comments and Suggestions for Authors: This manuscript needs a very deep revision to be reconsidered: Minor revision:

any imprecisions as for example line 67-68: “The aim of this study was to analyze the combined effect of DG along with ATV or EZT”, but authors analyze the effect of the single drugs and various combinations among them. Correct also legends adding the effect of combinations of drugs.

1. line 78, 0.5% or 5% as then reported in line 80? We have modified this part. (pag. 2; line 80)

2. Line 147, add” or normal diet”. Corrected

3.Table 2, “a” underlined, what does it mean? Or”,” before a letter? We have changed a for * and b for # in all manuscript tables.

Major revision:

Several mistakes about statistics if you consider what is written in the text respect to what is reported in tables. This makes very difficult to evaluate this manuscript. Just some example:

4. line 114, for HD+EZT group no statistics is reported in table 1. Corrected (pag. 3; table 1)

5. line 116, (p<0.0001, p<0.05 and p<0.0001; respectively) data not correct if I look at the table (b for all the groups)? We have changed (p<0.0001, p<0.05 and p<0.0001; respectively) for (p<0.05) in all manuscript.

6. line 128, p<0.001 or 0.0001? it is not written in table what value is “a”; somewhere in the text it is 0.0001. Corrected. We have improved the results section

7. line 177, (p< 0.05) but in table there it is not “b”. Done

-          etc, etc, etc…….

About Discussion:

8. too long because authors report several information which should be in introduction, not in this section. We have improved the discussion section

9. line 213, authors refer to fecal excretion in a wrong context (Biochemical parameters) and they talk about stimulation by diosgenin of cholesterol excretion but in their results I don’t find this (table 4). we correct this paragraph (pag. 6; line 207-211)

10.There are a lot of data not discussed. For example, authors don’t discuss epididymis values that in some cases seem statistically significant. Then, line 229, “HD rats had 50% more activity in the liver, kidney and spinal cord than the same tissues in ND” …kidney? It doesn’t seem from table 2. We have modified this part and the results in epididymis and kidney were commented in the discussion (pag. 7; line 238-251)

11.Then, I would compare data obtained for not labeled and labeled cholesterol that means discuss about the effect of drugs on cholesterol coming from synthesis and absorption (not labeled) with respect to the effect on cholesterol from just diet(labeled). For example, line 232-233, “DG diminished 26-14C-cholesterol activity in the liver and spinal cord”, compare this with data in table 1 and you will not find the same, it would be nice to discuss this. Moreover, data from spinal cord is not statistically relevant (see table 2). We have improved the discussion section.

12. Line 234 “ATV increased 26-14C-cholesterol activity detected in spinal cord”, please discuss why, in your opinion it is not the same for the combination ATV-EZT, considering also the effect of EZT alone. We have included in the discussion section (page 7; line: 237 and 238).

However, ATV significantly increases spinal cord activity possibly due to modification in the expression of the genes mentioned in the discussion section. Analyzing the results, we observed that the group with EZT has a similar effect on the kidney; the effect of certain drugs on the expression of proteins is known, in the case for example of CYP genes it can be observed that one substance increases its expression and another decreases it. Considering that effect and since there is no work that studies that effect in relation to ATV, EZT and cholesterol transport, we consider that when combining ATV with EZT somehow this effect is lost on the transporters of cholesterol in the spinal cord and the kidney, however further studies are needed in order to answer this question more appropriately. We appreciate your observation that helps enrich future goals in the study of cholesterol metabolism.

13. Line 252, “we observed increased fecal excretion of neutral sterols in DG treated rats”, it doesn’t seem from table 4. Corrected.  We referred to the increase in the concentration of labelled taurocholic acid in the intestine of DG rats.

Best regards

Juan C. Diaz-Zagoya

Round 2

Reviewer 2 Report

I would suggest another deep reading of the manuscript cause of some mistakes. As example: Line 195 EZ instead of EZT; sentence starting at line 217 is not clear; Sentence starting at line 220  is not clear; line 229 “distribution… of”, etc etc

In Results section:

line 117: (p<0.05)  is referred just to HD-DG treatment because in table 1 I don’t find # for HD-EZT

line 140: dpm/g  or 0.5g?

line 159: “In HD+ATV+EZT the kidney 14C activity surpassed that  of the liver (p<0.05)” but I don’t find any symbol of statistics in table 2 for HD+ATV+EZT kidney

line 179: “ higher in HD+DG and HD+ATV+DG (p<0.05)” but I don’t find any symbol of statistics in table 4 referred to these conditions

line 183: “but HD+DG, in which the activity was significantly lower” but I don’t find any symbol of statistics in table 4 referred to this condition.